# Predictors of Response to Exclusive Enteral Nutrition in Newly Diagnosed Crohn´s Disease in Children: PRESENCE Study from SEGHNP

**DOI:** 10.3390/nu12041012

**Published:** 2020-04-07

**Authors:** Melinda Moriczi, Gemma Pujol-Muncunill, Rafael Martín-Masot, Santiago Jiménez Treviño, Oscar Segarra Cantón, Carlos Ochoa Sangrador, Luis Peña Quintana, Daniel González Santana, Alejandro Rodríguez Martínez, Antonio Rosell Camps, Honorio Armas, Josefa Barrio, Rafael González de Caldas, Mónica Rodríguez Salas, Elena Balmaseda Serrano, Ester Donat Aliaga, Andrés Bodas Pinedo, Esther Vaquero Sosa, Raquel Vecino López, Alfonso Solar Boga, Ana Moreno Álvarez, César Sánchez Sánchez, Mar Tolín Hernani, Carolina Gutiérrez Junquera, Nazareth Martinón Torres, María Rosaura Leis Trabazo, Francisco Javier Eizaguirre, Mónica García Peris, Enrique Medina Benítez, Beatriz Fernández Caamaño, Ana María Vegas Álvarez, Laura Crespo Valderrábano, Carmen Alonso Vicente, Javier Rubio Santiago, Rafael Galera-Martínez, Ruth García-Romero, Ignacio Ros Arnal, Santiago Fernández Cebrián, Helena Lorenzo Garrido, Javier Francisco Viada Bris, Marta Velasco Rodríguez-Belvis, Juan Manuel Bartolomé Porro, Miriam Blanco Rodríguez, Patricia Barros García, Gonzalo Botija, Francisco José Chicano Marín, Enrique La Orden Izquierdo, Elena Crehuá-Gaudiza, Víctor Manuel Navas-López, Javier Martín-de-Carpi

**Affiliations:** 1Paediatric Gastroenterology and Nutrition Unit, Hospital Complex Torrecárdenas, 04009 Almeria, Spain; melindamoriczi@gmail.com (M.M.); galeramartinez@gmail.com (R.G.-M.); 2Department of Paediatric Gastroenterology, Hepatology and Nutrition, Hospital Sant Joan de Déu, Esplugues del Llobregat, 08950 Barcelona, Spain; gpujol@sjdhospitalbarcelona.org (G.P.-M.); javiermartin@sjdhospitalbarcelona.org (J.M.-d.-C.); 3Paediatric Gastroenterology and Nutrition Unit, Regional University Hospital of Malaga, 29011 Málaga, Spain; rafammgr@gmail.com; 4Gastroenterology Unit, Hepatology and Nutrition, Central University Hospital of Asturias, 33011 Oviedo, Spain; principevegeta@hotmail.com; 5Paediatric Gastroenterology, Hepatology and Nutrition Unit, University Hospital Vall d’Hebron, 08035 Barcelona, Spain; osegarra@vhebron.net; 6Paediatric Gastroenterology, Hepatology and Nutrition Unit, Hospital Virgen de la Concha, 49022 Zamora, Spain; cochoas2@gmail.com; 7Paediatric Gastroenterology, Hepatology and Nutrition Unit, Maternal-Child Hospital, 35016 Las Palmas of Gran Canaria, Spain; lpena@dcc.ulpgc.es (L.P.Q.); lpadanielgs@gmail.com (D.G.S.); 8Paediatric Gastroenterology, Hepatology and Nutrition Unit, Hospital Virgen del Rocío, 41013 Seville, Spain; ARMGASTROPEDIATRIA@gmail.com; 9Paediatric Gastroenterology and Nutrition Unit, University Hospital Son Espases, 07010 Palma de Mallorca, Spain; antoniorosellcamps@gmail.com; 10Paediatric Gastroenterology, Hepatology and Nutrition Unit, University Hospital of the Canary Islands, 38320 Tenerife, Spain; armashonorio9@gmail.com; 11Paediatric Gastroenterology, Hepatology and Nutrition Unit, University Hospital Fuenlabrada, Fuenlabrada, 28942 Madrid, Spain; josefa.barrio@salud.madrid.org; 12Paediatric Gastroenterology, Hepatology and Nutrition Unit, Hospital Reina Sofía, 14004 Cordoba, Spain; rgonzalezdecaldasmarchal@gmail.com (R.G.d.C.); salas002@hotmail.com (M.R.S.); 13Paediatric Gastroenterology, Hepatology and Nutrition Unit, Hospital Complex, University of Albacete, 02008 Albacete, Spain; elenabalmaseda@hotmail.com; 14Paediatric Gastroenterology, Hepatology and Nutrition Unit, University Hospital La Fe, 46009 Valencia, Spain; esterdonat@gmail.com; 15Paediatric Gastroenterology, Hepatology and Nutrition Unit, Hospital Clinic San Carlos, 28040 Madrid, Spain; andresbpinedo@yahoo.es (A.B.P.); estvaqsos@yahoo.es (E.V.S.); rvecinolopez@gmail.com (R.V.L.); 16Paediatric Gastroenterology, Hepatology and Nutrition Unit, Maternal-Child Hospital Teresa Herrera, 15004 A Coruña, Spain; Alfonso.Solar.Boga@sergas.es (A.S.B.); Ana.Moreno.Alvarez@sergas.es (A.M.Á.); 17Paediatric Gastroenterology, Hepatology and Nutrition Unit, General University Hospital Gregorio Marañón, 28009 Madrid, Spain; sangoros@hotmail.com (C.S.S.); marth81@gmail.com (M.T.H.); 18Paediatric Gastroenterology, Hepatology and Nutrition Unit, University Hospital Puerta de Hierro, 28220 Majadahonda, Madrid, Spain; carolinagjunquera@gmail.com; 19Paediatric Gastroenterology, Hepatology and Nutrition Unit, University Hospital Clinic of Santiago, 15706 Santiago de Compostela, Spain; nazarethmt@hotmail.com (N.M.T.); mariarosaura.leis@usc.es (M.R.L.T.); 20Paediatric Gastroenterology, Hepatology and Nutrition Unit, University Hospital of Donostia, 20006 San Sebastian, Spain; JAVIEREIZAGUIRRE@hotmail.com; 21Paediatric Gastroenterology, Hepatology and Nutrition Unit, Hospital Lluís Alcanyís, Xátiva, 46800 Valencia, Spain; monika_gp@hotmail.com; 22Paediatric Gastroenterology, Hepatology and Nutrition Unit, Hospital 12 de Octubre, 28041 Madrid, Spain; enrique.medina@salud.madrid.org; 23Paediatric Gastroenterology, Hepatology and Nutrition Unit, Hospital Álvaro Cunqueiro, 36213 Vigo, Spain; BEATRIZFDEZCAAMANO@gmail.com; 24Paediatric Gastroenterology, Hepatology and Nutrition Unit, Hospital Río Hortega, 47012 Valladolid, Spain; ANAVEGAS@telefonica.net (A.M.V.Á.); lcrespova@saludcastillayleon.es (L.C.V.); carmenalonso@gmail.com (C.A.V.); 25Paediatric Gastroenterology, Hepatology and Nutrition Unit, Hospital de Jerez, 11408 Jerez de la Frontera, Cádiz, Spain; rubiosantiago.javier@gmail.com; 26Paediatric Gastroenterology, Hepatology and Nutrition Unit, Paediatric Hospital Miguel Servet, 50009 Zaragoza, Spain; ruthgarciaromero@yahoo.es (R.G.-R.); rosnacho@yahoo.es (I.R.A.); 27Paediatric Gastroenterology, Hepatology and Nutrition Unit, University Hospital Complex Ourense, 32616 Ourense, Spain; sfcebrian@gmail.com; 28Paediatric Gastroenterology, Hepatology and Nutrition Unit, Hospital Basurto, Bilbao, 48013 Vizcaya, Spain; HELENA.LORENZOGARRIDO@osakidetza.eus; 29Paediatric Gastroenterology, Hepatology and Nutrition Unit, University Hospital Niño Jesús, 28009 Madrid, Spain; ELVYTTO@gmail.com (J.F.V.B.); martavrb@gmail.com (M.V.R.-B.); 30Paediatric Gastroenterology, Hepatology and Nutrition Unit, Hospital Río Carrión, 34005 Palencia, Spain; jumabarpo@hotmail.com; 31Paediatric Gastroenterology, Hepatology and Nutrition Unit, Jiménez Díaz Foundation, 28040 Madrid, Spain; mblanco@fjd.es; 32Paediatric Gastroenterology, Hepatology and Nutrition Unit, Hospital San Pedro de Alcántara, 10003 Cáceres, Spain; pbarrosg@yahoo.es; 33Paediatric Gastroenterology, Hepatology and Nutrition Unit, Alcorcón Foundation, 28922 Alcorcón, Madrid, Spain; g_botija@yahoo.es; 34Paediatric Gastroenterology, Hepatology and Nutrition Unit, University Hospital Los Arcos del Mar Menor, 30739 Pozo Aledo, Murcia, Spain; fjchicano@ono.com; 35Paediatric Gastroenterology, Hepatology and Nutrition Unit, University Hospital Infanta Sofía, 28709 San Sebastián de los Reyes, Madrid, Spain; enrique.orden@quironsalud.es; 36Paediatric Gastroenterology, Hepatology and Nutrition Unit, University Clinical Hospital, 46010 Valencia, Spain; elenacrehua@gmail.com; 37Institute of Biomedical Research in Malaga (IBIMA), 29010 Málaga, Spain

**Keywords:** inflammatory bowel disease, Crohn’s disease, exclusive enteral nutrition, calprotectin, children, paediatric

## Abstract

Exclusive enteral nutrition (EEN) has been shown to be more effective than corticosteroids in achieving mucosal healing in children with Crohn´s disease (CD) without the adverse effects of these drugs. The aims of this study were to determine the efficacy of EEN in terms of inducing clinical remission in children newly diagnosed with CD, to describe the predictive factors of response to EEN and the need for treatment with biological agents during the first 12 months of the disease. We conducted an observational retrospective multicentre study that included paediatric patients newly diagnosed with CD between 2014–2016 who underwent EEN. Two hundred and twenty-two patients (140 males) from 35 paediatric centres were included, with a mean age at diagnosis of 11.6 ± 2.5 years. The median EEN duration was 8 weeks (IQR 6.6–8.5), and 184 of the patients (83%) achieved clinical remission (weighted paediatric Crohn’s Disease activity index [wPCDAI] < 12.5). Faecal calprotectin (FC) levels (μg/g) decreased significantly after EEN (830 [IQR 500–1800] to 256 [IQR 120–585] *p* < 0.0001). Patients with wPCDAI ≤ 57.5, FC < 500 μg/g, CRP >15 mg/L and ileal involvement tended to respond better to EEN. EEN administered for 6–8 weeks is effective for inducing clinical remission. Due to the high response rate in our series, EEN should be used as the first-line therapy in luminal paediatric Crohn’s disease regardless of the location of disease and disease activity.

## 1. Background

Exclusive enteral nutrition (EEN) is the use of a complete liquid formula as the sole source of food for 6 to 8 weeks. It is still the therapeutic modality of choice for treating the first flare-up of paediatric luminal Crohn’s disease (CD) [1]. Although the efficacy of EEN in inducing remission has been known since the 1980s, there have been few studies published on the subject in Spain [2]. In 2014, the Inflammatory Bowel Disease (IBD) working group of the Spanish Society of Gastroenterology, Hepatology, and Nutrition (SEGHNP) published the PRESENT (PREScription of Enteral Nutrition in pediaTric Crohn’s disease in Spain) survey (70-item questionnaire) [3]. The aim of the PRESENT study was to investigate the frequency and characteristics of the use of EEN, including barriers and enablers, in pediatric gastroenterology units in Spain. The PRESENT study revealed that many paediatric gastroenterologists limited their indication of EEN as first-line therapy at the onset of the disease to specific cases. In this way, 43% of the physicians only indicated EEN for inflammatory forms (B1), 37.3% indicated EEN only when there was ileal (L1) or ileocolonic (L3) involvement, 41.1% only indicated EEN for mild-moderate disease, 62.7% only indicated EEN if the patient and their family were cooperative, and only 25.5% offered EEN as the only therapeutic option (they allowed them to choose between ENN and steroids). From this study, it appears that a group of newly diagnosed Crohn’s disease children, according to their phenotypic characteristics and severity of the disease (measured by Paediatric Crohn’s Disease Activity Index, PCDAI), would not benefit from receiving EEN as first-line therapy, as they were not considered, a priori, a candidate by their treating physician.

Our working hypothesis is that the response to EEN in patients with Crohn’s disease does not depend on the severity of the flare-up measured by weighted Pediatric Crohn’s Disease Activity Index (wPCDAI) [4], on the patient’s age, or on the location of the disease (according to Paris classification [5]). There are other factors, probably still unknown, that limit the success of this therapeutic modality. The aims of the present study were to determine the rate of remission after induction to remission therapy with EEN in newly diagnosed Crohn’s disease children, the potential predictors of response to EEN and the need for treatment with biological agents during the first 12 months of the disease. 

## 2. Material and Methods

We conducted a retrospective cohort study that included paediatric patients diagnosed with CD between 1 January 2014 and 31 December 2016, based on their clinical, laboratory, endoscopic, radiological, and histological criteria, according to ESPGHAN Revised Porto Criteria for the Diagnosis of Inflammatory Bowel Disease in Children and Adolescents [6], and who were treated with EEN for their first flare-up and were followed up for 12 months after diagnosis. The participating centres were invited through the distribution list of the Spanish Society of Gastroenterology, Hepatology, and Nutrition (*Sociedad Española de Gastroenterología, Hepatología y Nutrición*, SEGHNP) in January 2017. A Case Report Form (CRF) was designed and distributed by email to the participating centers. The deadline for sending the CRFs to the study coordinator was 28 February 2018. All patients received one of the commercially prepared formulas (lactose and gluten free), normal or hypercaloric polymeric flavored formulas (ready for drink) or Modulen IBD^®^ or Resource IBD^®^ (the same formula but with a different name, Nestlé Health Science, Vevey, Switzerland), both prepared the same way by mixing 1700 ml water with 400 g of product to produce 2000 ml of formula (1 kcal/ml). No other food was allowed during the EEN period. The prescribed volumes were based on the caloric needs of each patient according to age, sex, and nutritional status. Feeds were gradually increased to target volumes in 3–5 days and patients were only allowed to drink water during treatment. EEN was given for a 6- to 8-week period. After the EEN period, food was introduced according to local protocol. 

The data collected were age, gender, type of delivery, breastfeeding, previous appendectomy, rural or urban (more than 10.000 population) environment, family history of IBD, season of the year in which it was diagnosed, time from the onset of symptoms to diagnosis, and concomitant pharmacological treatment. Disease phenotype was determined according to the Paris classification [5]. 

All the patients were assessed at the start and after completing the EEN period, and the variables analyzed were weight, height, body mass index (BMI), C-reactive protein (CRP), erythrocyte sedimentation rate (ESR), albumin, complete blood count (CBC), faecal calprotectin (FC), and wPCDAI [4]. The wPCDAI was derived by reweighting the PCDAI mathematically, and composed of three domains—clinical history symptoms with a 1 week recall (abdominal pain, patient functioning, general well-being, and stools per day), physical examination (weight, perirectal disease, and extraintestinal manifestations), and laboratory parameters (ESR and albumin) with individual items mathematically weighted to produce an overall score that classifies patients into four disease activity categories: none, < 12.5; mild, 12.5 to 40; moderate, >40 to 57.5; and severe, >57.5. Weight and height were measured with the patient barefoot and in underwear. Weight, height, and BMI z scores were calculated using data from Spanish growth charts [7]. Samples for the determination of FC were collected at home by the patient the day before and were delivered, refrigerated, to the laboratory for immediate analysis. Blood samples were collected at each participating hospital and processed on site. There was no centralized analysis of blood or faecal samples. We considered remission a wPCDAI <12.5 points after completing the EEN period and defined a response to EEN as a >17.5-point change in the initial wPCDAI value without achieving remission. Biological remission was defined as wPCDAI <12.5, ESR <20 mm/h, CRP <5 mg/L, and FC <300 μg/g. We excluded patients with ulcerative colitis (UC), those with inflammatory bowel disease unclassified (IBD-U), those undergoing concomitant treatment with steroids or anti-tumour necrosis factor (anti-TNF) during induction with EEN, and those who were treated with EEN in successive flare-ups. Written informed consent was obtained from parents and also from patients older than 12 years old before collecting the data on the CRF. 

### 2.1. Statistical Analysis

Variables with a normal distribution were expressed as mean ± standard deviation, and those without a normal distribution as median and interquartile range (IQR). We employed the Kolmogorov–Smirnov test to evaluate the normality of the distribution. We employed the Wilcoxon signed-rank test for paired samples (wPCDAI, CRP, FC, ESR, albumin, and haemoglobin values before and after EEN) and the chi-square test for comparing the proportions (A1b of Paris, wPCDAI ≤ 57.5, Ileal involvement, CRP >15 mg/L and FC <500 μg/g). To compare the variables with a normal distribution, we employed the t-Student (age at diagnosis in responders and non-responders) and the Mann–Whitney U test in those without normal distribution (time to diagnosis, months; wPCDAI; CRP, mg/L and FC, μg/g values; and time to anti-TNF between responders and non-responders; EEN duration between those who responded and those who went into remission after EEN). We estimated the hazard ratio and its confidence interval using the Cox proportional hazards model. We considered a *p* < 0.05 as statistically significant. We constructed predictive models using univariate and multivariate logistic regression tests. To construct the model, all the variables included on Table 2 were included first in the univariate analysis, and only those variables that presented statistically significant differences or a trend (*p* < 0.15) in the univariate analysis, along with the variables that, based on the theoretical or empirical knowledge, were considered related to the dependent variable and were included on the MV on Table 3. We measured the magnitude of the association between the model’s predictive variables and the dependent variable with the odds ratio (OR) and its corresponding 95% confidence interval (CI). Accepting an alpha risk of 0.05 in a two-sided test with 1307 participants in the first group (2) and 222 in the second, a statistical power of 85% was needed to recognize as statistically significant the difference between 0.74 in the first group and 0.83 in the second group.

### 2.2. Ethical Issues

The study (Code 0623-M2-14) and protocols for recruitment were initially approved by the Ethics Committee of the Hospital Regional Universitario de Málaga. Later it was approved by the rest of the ethics committees of the participating centers.

## 3. Results

We received data from 235 patients; of these, we analysed a total of 222 patients from 35 hospital centres and discarded 13 due to a lack of availability of all requested information. Only 11.2% (25/222) of the patients came from rural areas, 37 (17%) patients had a family history of IBD (14% CD, 2.5% UC and 0.5% IBD), 29% were born by caesarean section, and 64% had been breastfed for a median of 3 months from birth. Some 26.7% had their initial presentation in winter, and only 3% had received an appendectomy. The time to diagnosis did not differ significantly between those who had a family history of IBD and those who did not (4.7 vs. 4.4 months, *p* = 0.977). The clinical characteristics are listed in Table 1. 

The median EEN duration for the entire series was 8 weeks (IQR 6.7–8.5) and was significantly longer for the patient group that achieved remission than for the groups that did not (8 [IQR 7.0–8.7] weeks for the remission group vs. 7.7 [IQR 4.6–8.1] weeks for the non-remission responders, *p* = 0.019]). The median time to response was 15 days (IQR 11–27). Of the 222 patients, 178 (80%) were administered Modulen IBD^®^/Resource IBD^®^ (Nestlé Health Science, Vevey, Switzerland), while the rest (20%) were administered other polymeric formulas; 18 (8.1%) patients required a nasogastric tube (NGT) to administer the enteral formula. By the end of the EEN period, 83% of the patients (184/222) had achieved clinical remission (ΔwPCDAI, −47 ± 18), and an additional 12.3% (27/222) responded but did not achieve clinical remission (ΔwPCDAI, −41 ± 16) (Figure 1a). A significant reduction in C-reactive protein (CRP), faecal calprotectin (FC) and erythrocyte sedimentation rate readings was also observed (Figure 1b–d), as well as a significant increase in albumin and haemoglobin values (Figure 1e,f). 

After the EEN period, all patients started on a normal diet. In addition, 82.9% of the patients who had not achieved clinical remission and 85.9% of those who had were supplemented every day with 500 ml of polymeric formula over the following months. Of the 38 (17.1%) patients who did not achieve clinical remission after EEN, nine had been administered steroids (eight received prednisone, and one budesonide) for 4 weeks (IQR 4–7.5) from the start of the EEN, and 29 had been administered anti-TNF (16 infliximab, and 23 adalimumab) for a median of 1.4 months (IQR 0–5) since EEN was initiated. The time to the start of anti-TNF treatment was longer for those who responded but did not achieve remission than for those who did not respond (3.2 months [IQR 1.4–7.4] vs. 0 months [IQR 0–1.2], *p* = 0.009). Of the nine patients who were started on steroids due to lack of response to EEN, five achieved clinical remission (55.5%), but all started treatment with anti-TNF during the first year of follow-up (8 months, IQR 5–10).

### 3.1. Predictors of Response to Exclusive Enteral Nutrition

Table 2 shows the differences in baseline clinical activity and laboratory biomarkers in responders (wPCDAI < 12.5) versus nonresponders. In terms of predictors of response (Table 3), those patients with a mild–moderate disease (wPCDAI < 57.5), with FC < 500 μg/g, ileal involvement, and CRP > 15 mg/L showed a better response to EEN. 

### 3.2. Time to Biological Therapy

At 12 months of follow-up, 84/222 (37%) of the patients had been started on therapy with anti-TNF (36 on infliximab and 48 on adalimumab) after a median of 5 months (IQR 1.7–8.7) from finishing EEN. The time taken to the start of anti-TNF therapy was longer for those with clinical remission (6.8 months [IQR 3.5–10.2] vs. 1.3 months [IQR 0–4.3], *p* < 0.0001) and with biological remission (7.1 months [IQR 4.1–10.5] vs. 3.4 months [IQR 0.6–8.2], *p* = 0.056). Clinical remission after EEN (wPCDAI < 12.5) (Figure 2A) and biological remission (wPCDAI < 12.5, ESR < 20 mm/h, CRP < 5 mg/L and FC 300 μg/g) (Figure 2B) were associated with a significantly reduced risk of starting anti-TNF therapy in the following 12 months. At the start of anti-TNF therapy, the patients’ mean wPCDAI, CRP, and FC were 33.7 (IQR 17.5–42.5), 13.8 (IQR 2.7–30.1), and 992 (IQR 376–2213), respectively.

## 4. Discussion

The present study reinforces the literature’s efficacy data for EEN in newly diagnosed CD children regardless of age. Our results for the clinical remission rates are similar to the previously published data [2]. EEN induces clinical remission [8,9] in the first flare-up or during relapses [10,11], induces mucosal [12] and transmural healing [13], has a positive effect on growth [14], on bone health [15], on the nutritional state [16], and on the health-related quality of life [17], and is a therapeutic option for decreasing the risk of relapses during follow-up [18], to avoid treatment with steroids [14] and to update these patients’ vaccination schedules before starting immunosuppressive therapy [19]. 

EEN should therefore be the first option when treating paediatric patients after diagnosing luminal CD, given its demonstrated effects in the short, medium, and long term (growth, bone health, and vaccinations) in the absence of the adverse effects of steroids. In our patient group, all those who started steroids after EEN failure required a shift to biological therapy during the first year of follow-up (8 months, IQR 5–10). It therefore makes sense not to subject patients to steroids and instead directly start therapy with anti-TNF. A recent clinical trial conducted with a paediatric population showed that the top-down (TD) strategy was superior to the step-up strategy for achieving endoscopic remission at week 10 and sustained clinical remission at 52 weeks without requiring other treatments or surgery [20]. This TD strategy is standard practice in Europe, the United States, and Canada [21]. A meta-analysis of studies conducted with adult populations revealed that the use of enteral nutrition in combination with infliximab was more effective for inducing and maintaining clinical remission among patients with CD than monotherapy with infliximab [22], which is based on an etiopathogenic standpoint [23] and by the capacity of EEN to change the proinflammatory cytokine profile in the intestinal mucosa [24]. This strategy, which was not evaluated in our study, could be assessed in an ad hoc designed study. 

Regarding the predictors to response, although our model is statistically significant, it only partially explains the dependent variable (Cox-Snell R^2^: 0.130 and Nagelkerke R^2^: 0.202), indicating that additional factors play a role in the response [25]. Unlike CRP, which is related to transmural involvement, FC is correlated with mucosal involvement [26]. In our case, CRP indicated the presence of an inflammatory pattern. Although EEN was initially thought to be more effective in patients with ileal involvement than in those with exclusively colonic involvement (L1 or L3 versus L2 according to Paris classification), there is no quality evidence to support this belief [27]. In our series, however, we did find that ileal involvement was a predictor of a response to EEN. This finding could be partly explained by the fact that the ileum is considered by a number of authors as the location from which all phenotypic forms of CD originate. The authors confer a predominant role to the ileal microbiota in the prognosis and treatment response [28]. Moreover, biological remission (PCDAI, CRP, and FC) correlates better with the degree of mucosal inflammation [29] than the routinely employed activity indices [4]. The growth study [30] revealed that biological remission at week 12 (PCDAI < 5, calprotectin < 400 μg/g and CRP < 20 mg/L) was the best predictor of relapses and complications during follow-up. In our case, biological remission (wPCDAI < 12.5, CRP < 5 mg/L, ESR < 20 mm/h and FC < 300 μg/g) after EEN predicted the need for anti-TNF therapy in the following 12 months. An important aspect here is the role of other drugs in maintaining the remission after induction therapy. In our series, more than 80% of the patients were treated with thiopurines concomitantly with the EEN. The results of the initial study by Markowitz et al. [31] were not reproduced in two multicentre studies conducted with adults [32,33]. A prospective cohort of patients with IBD showed a modest role for thiopurines in maintaining remission [34]. Similarly, a considerable majority of patients were supplemented with a polymeric formula once the EEN period had been completed, although the evidence regarding the effectiveness of the polymeric formula in this setting is scarce [35]. There were no complications resulting from the use of EEN. 

In our study, 40% of the patients were started on anti-TNF therapy during the first year from the diagnosis, compared with 70% in the series published by Jongsma et al. [20]. This difference, in addition to being affected by the design of the two studies, could be due to the fact that despite the current tendency to start anti-TNF therapy earlier [36], there is still a certain reluctance in our setting to use anti-TNF in the initial phases of the disease, after the failure of induction with EEN.

One of the limitations of EEN is willingness on the part of the child and their family. Compliance with EEN was high in our study, unlike that reported in other studies, which attributed the low compliance to problems in the formulas’ taste and to monotony [37]. In our series, less than 10% of patients needed NGT (18 out of 222) and none of them declined the EEN, except for medical indication due to the lack of response. The development of the CD Exclusion Diet (CDED), which partly minimises the problems of EEN, has enabled a turnaround in the treatment of paediatric CD. The CDED has been shown to respond to a frequent demand by patients and their relatives regarding acceptability and compliance [38]. The CDED is as effective as EEN in achieving clinical and biochemical remission and mucosal healing but superior to EEN in tolerance and compliance. CDED, unlike EEN, constitutes a long-term strategy for maintaining remission and is nutritionally balanced. By including fibre, CDED corrects the bacterial dysbiosis present in these patients [39], and is therefore a much more realistic and advanced approach than EEN if complied with adequately. 

The most relevant limitations of the present study are its retrospective and multicentered nature, which limits the quality of the data and their corresponding analysis. Another issue to consider is the inclusion bias, given that the researchers decided whether or not the patients were candidates for EEN. Moreover, the small sample size in some of the subgroups could explain the considerable variation in the confidence intervals in a number of the measurements. The results should therefore be interpreted with caution. 

## 5. Conclusions

In conclusion, EEN administered for 6–8 weeks is effective for inducing clinical remission. Given the high response rate in our series, EEN should be employed as the first-line therapy for luminal paediatric CD, regardless of the location of the disease, the CRP and FC levels, and the wPCDAI. Some patients, however, will respond better than others to EEN. 

## Figures and Tables

**Figure 1 nutrients-12-01012-f001:**
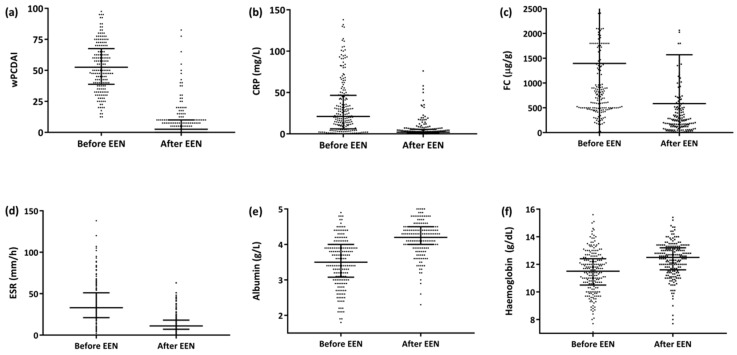
Progression of the activity index and biochemical parameters after the EEN period (*n* = 222). (**a**) wPCDAI [52.5 (IQR 39–67) vs. 2.5 (IQR 0–10), (ΔwPCDAI −44 ± 20 *p* < 0.0001]. (**b**) CRP [21 (IQR 6–47) vs. 2 (IQR 1–5), *p* < 0.0001]. (**c**) FC [830 (IQR 500-1800) vs. 256 (IQR 120–585), *p* < 0.0001]. (**d**) SS [33 (IQR 21–51) vs. 11 (IQR 7–18), *p* < 0.0001]. (**e**) Albumin [3.5 (IQR 3–4) vs. 4.2 (IQR 4–4.5), *p* < 0.0001. (**f**) Haemoglobin (11.5 [IQR 10.5–12.4] vs. 12.5 [IQR 11.6–13.2], *p* = 0.0001). Abbreviations: CRP, C-reactive protein; FC, faecal calprotectin; wPCDAI, weighted Paediatric Crohn’s Disease Activity Index; EEN: exclusive enteral nutrition.

**Figure 2 nutrients-12-01012-f002:**
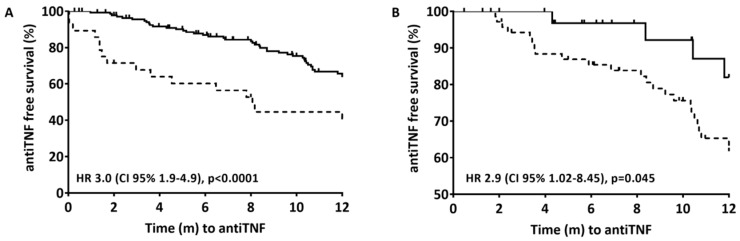
Time from the end of induction with EEN to the start of anti-TNF therapy during the first 12 months after diagnosis. Cox proportional hazard models; only the stratification variables were included (clinical remission in 2A and biological remission in 2B). (**A**) Solid line is clinical remission (wPCDAI <12.5) after EEN; Dashed line is no clinical remission after EEN. Stratified by clinical remission (6.8 months [IRQ 3.5–10.2] vs 1.3 months [IQR 0–4.3], *p* < 0.0001). (**B**) Solid line is biological remission (wPCDAI <12.5, ESR <20 mm/h, CRP <5 mg/L and FC <300 μg/g); Dashed line is clinical remission (wPCDAI <12.5). Stratified by biological remission (7.1 months [IQR 4.1–10.5] vs 3.4 months [IQR 0.6–8.2], *p* = 0.056).

**Table 1 nutrients-12-01012-t001:** Baseline characteristics of the patients treated with exclusive enteral nutrition (*n* = 222).

**Gender, males (%)**	140 (63.1%)
**Age at diagnosis, years**	11.6 ± 2.50
**Time from onset of symptoms to diagnosis, months**	4.4 (IQR 2.5–9.8)
**Anthropometric measurements at diagnosis ^A^**	
Weight, kg [z score]	36.1 ± 11.8 [-0.79 ± 0.98]
Height, cm [z score]	145.5 ± 17.5 [0.34 ± 0.4]
BMI, kg/m^2^ [z score]	16.4 ± 2.8 [-0.76 ± 0.9]
**Paris classification**	**n (%)**
L1	42 (18.9)
L2	22 (9.9)
L3	82 (36.9)
L4a	18 (8.1)
L4b	2 (0.9)
L3L4a	46 (20.7)
L3L4b	8 (3.6)
L3L4ab (extensive)	2 (0.9)
B1	216 (97)
B2	6 (3)
Perianal disease (p)	26 (12)
Growth retardation (G1)	49 (22)
**Baseline wPCDAI ^B^**	52.5 (IQR 39–67)
Mild	63 (28.5%)
Moderate	74 (33.0%)
Severe	85 (38.5%)
**Concomitant treatment with EEN**	**n (%)**
5-ASA	42 (19)
Thiopurines (AZA/6MP)	191(86)
Antibiotics (Metronidazole/Azithromycin)	47 (21)
**Baseline laboratory results**	
Faecal calprotectin, μg/g	830 (IQR 500–1800)
CRP, mg/L	21.1 (IQR 6–47)
ESR, mm/h	33 (IQR 21–51)
Albumin, g/dL	3.5 (IQR 3.0–4.0)
Vitamin D, ng/mL	24 (IQR 19–30)
Hb, g/dL	11.5 (IQR 10.5–12.4)
Htc, %	36 (IQR 33–38.5)
WBC, x10^9^/L	9.7 ± 4.3
Platelets, x10^9^/L	467 ± 150

^A^ Reference values [7]. ^B^ wPCDAI: weighted Paediatric Crohn’s Disease Activity Index: Remission < 12.5; Mild 12.5–40; Moderate > 40; Severe > 57.5 points. Adapted from Turner et al. [4] Abbreviation: 6MP, 6 mercaptopurine; AZA, azathioprine; CRP, C-reactive protein; ESR, erythrocyte sedimentation rate; Hb, haemoglobin; Hct, haematocrit; IQR, interquartile range; WBC, white blood cells. BMI: Body Mass Index. Paris classification adapted from Levine et al. [5]: L1: distal 1/3 ileal ± limited cecal disease; L2: colonic; L3: ileocolonic; L4a: upper disease proximal to ligament Treitz; L4b: upper disease distal to ligament of Treitz and proximal to distal 1/3 ileum. B1: non-stricturing non-penetrating. B2: stricturing.

**Table 2 nutrients-12-01012-t002:** Baseline clinical activity and laboratory biomarkers in both groups: Responders (wPCDAI < 12.5) and Nonresponders.

Variable	Responders (*n* = 184)	Nonresponders (*n* = 38)	*p*
A1b of Paris	41 (22.2%)	8 (21.0%)	0.485 ^#^
Age at diagnosis, years	12.1 ± 2.4	11.4 ± 2.9	0.637 ^&^
Time to diagnosis, months	4.3 (IQR 2.5–10.0)	4.6 (IQR 2.5–9.6)	0.877 *
wPCDAI	50 (IQR 37.5–65)	60 (IQR 47.5–72.5)	0.015 *
wPCDAI ≤ 57.5	122 (66.3%)	16 (42.1%)	0.011 ^#^
Ileal involvement	180 (97.8%)	35 (92.1%	0.064 ^#^
CRP, mg/L	22.5 (IQR 8.9–47)	18.4 (IQR 5.9–59.5)	0.851 *
CRP >15 mg/L	117 (63.6%)	21 (55.2%)	0.201 ^#^
FC, μg/g	797 (IQR 492–1800)	1285 (IQR 759–2750)	0.011 *
FC < 500 μg/g	47 (25.5%)	2 (5.2%)	0.006 ^#^

Abbreviations: CRP, C-reactive protein; EEN, exclusive enteral nutrition; FC, faecal calprotectin; IQR, interquartile range; wPCDAI, weighted Paediatric Crohn’s Disease Activity Index. ^#^ Chi-square test; * Mann-Whitney U test; ^&^ t-Student test.

**Table 3 nutrients-12-01012-t003:** Predictive variables of response to EEN. Multivariate analysis. Dependent variable: wPCDAI < 12.5. *n* = 222 patients.

Variable	Univariate OR (CI 95%)	*p*	Multivariate OR (CI 95%)	*p*
wPCDAI ≤ 57.5	2.7 (1.2-6.0)	0.013	3.8 (1.5–9.7)	0.005
FC <500 μg/g	5.5 (1.2–24.3)	0.022	6.9 (1.3–35.4)	0.019
CRP >15 mg/L	1.4 (0.7–3.0)	0.306	2.6 (1.01–6.8)	0.047
Ileal involvement	5.1 (1.0–26.5)	0.049	6.3 (1.09–36.6)	0.039

Hosmer and Lemeshow test: *p* = 0.962; Cox-Snell R^2^: 0.130. Nagelkerke R^2^: 0.202; Sensitivity: 96 (91–99); Specificity 23 (10–42); PPV: 82 (75–89); NPV: 64 (31–89). Note: This table only shows the results of the univariate analysis of the variables that were finally included in the multivariate analysis. The model displayed here is significant, explains between 0.130 and 0.202 of the dependent variable, and correctly classifies 81% of cases.

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
