# Peer review of "Predictors of Response to Exclusive Enteral Nutrition in Newly Diagnosed Crohn´s Disease in Children: PRESENCE Study from SEGHNP"

_nutrients, 2020, doi:10.3390/nu12041012_

Round 1

Reviewer 1 Report

The paper attempts to assess the link between EEN and rate to remission of CD in pediatric population, and the need for additional therapy (after EEN). This is an interesting and worthy questions considering the burden of CD, and the possible side effect of current alternatives to EEN. Unfortunately the writing style makes it very difficult to adequately understand and interpret the content of the paper. The introduction and methods sections are woefully inadequate, and that makes  it very difficulty to critically review the results. The results seem interesting but the lack of information on how the various  data was generated makes the interpretation difficult. The authors provided a lot of background in the discussion, which was good. However, they should have committed more time to actually discussing the results presented in the paper.

Introduction:

This is very poorly written. The limited information provided in the first paragraph does not support  the working hypothesis and the specific goals of the paper.

First paragraph: Add a sentence or two to explain what Exclusive Enteral Nutrition means.

The introduction needs to be substantially revised to include the following:

  1. Explain EEN to your readers.
  2. Provide a basis for the working hypothesis. Present a justification of why the authors think that response to EEN  is independent of disease severity, age or the extent of the disease? How is severity of flare-up different from extent of the disease?
  3. Line 54: Provide more information on the clinical features of the patient group which did not respond to
  4. It is confusing to have a working hypothesis which is very different from the study aim.
  5. Revise the introduction to clearly highlight evidence on the factors which predict the benefits (or lack of benefits) of EEN.

Materials and Methods:

This section is also very inadequately written. Several important information were not presented. Hence it is impossible to understand  the method of data collection.

Provide enough information to help the reader understand why you are referring to this study as a retrospective cohort.  Just because you are using pre-exiting data does not necessarily make the study retrospective.

Indicate whether this was a rolling enrollment. When was the first and last persons enrolled?

How long was the follow-up.

Provide a detailed description of the EEN protocol used in this  project

Provide a detail description of the study/patient population.

Describe clearly the inclusion and exclusion criteria

Describe in detail the exact data/samples collected from study participant.

How were bio samples, stored and analyzed and in which labs?

Provide a description of the informed consent procedures

Ethical issue.

Provide the official names of all institutions which provided ethical clearance for this study

Statistical analyses:

The statistical analyses is not at all specific. The authors provided a general description of various statistical tests, without indicating which questions were addressed with the test. This section should be revised to clearly indicate the questions being addressed with each test, and the variables involved. Include a description of  how potential confounding was addressed in this paper.

Provide a clear explanation of  how  wPCDAI was calculated and how it should be interpreted.

How was first flare up  assessed and defined?

Results

The columns in table 1 should have headings.

All abbreviations in  Table 1 should be  defined in the footnote

All abbreviations in  Figure 1 should be  defined in the footnote

How was IBD diagnosed/defined?

In Table 2, are the  P-value generated from uni-variate or multivariate models. If they are from univariate model, also include additional columns  to include results from a mulitivariate model.

In table 3, what other variables were included in the model. Include the results for the variables that were not statistically significant, so that your readers can fully interpret the data.

Discussion:

The authors provided important information in this section to enable a broader understanding of the  problem of pediatric CD. The authors also discussed the acceptability of EEN and possible alternatives to EEN.

The authors spent more time discussing the literature than discussing the actual results presented in the paper.  The authors did not adequately discuss why some variables (but not others) were predictors. It appears from table 1 that  EEN is not beneficial to the  most severe cases (based on WPCDAI and FC values) but  this was not discussed.

The authors mentioned age in their hypothesis, but this was not a focus of their results and discussion.

Reviewer 2 Report

I read with interest the MS "PREdictors of reSponse to exclusive ENteral nutrition 2 in newly diagnosed Crohn´s disease childrEn: 3 PRESENCE Study from SEGHNP" by Moriczi M and coworkers. It is a well designed study dealing with nutritional treatment approach to IBD. The Authors should be commended for trying to implement the non-biological approach to IBD Tx in such a thoughtful way. I have just a minor suggestions: I could not find the clinical picture of those starting biologics in the 12 months following enteral nutrition. I believe it would be of major help to learn whether extensive bowel involvement and/or lenght of disease influenced the needing to start biologics, to quote some potential factors. I trust the Authors could give us additional relevant data on the issue. No additional points on this side.

Reviewer 3 Report

The study is important and relevant as it evaluates clinical practice. I have a few small comments. In the abstract I would change the last sentence to '..EEN should be used as the first-line therapy in luminal paediatric Crohn's disease regardless of the location of disease and disease activity'. In line 61 describing 'aims' I would change 'the predictors' to 'potential predictors'. In the statistics chapter I would like an explanation to the choice of p<0.15 in the univariate analyses to include a variable in the multiivariate analyses, as this is not related to significance but a selection criteria for making sure all relevant variables are included. In line 96 a reference number for the Ethics approval may be added. In the discussion the finding that longer treatment with EEN gave better results may be expanded on a little further. In line 133 I would add the percentage to 38 patients in brackets. In line 209 PCR should be changed to CRP. Also the finding that compliance to EEN was good may be expanded on as this is very interesting. All in all an excellent paper with a balanced discussion of clinical importance.

Round 2

Reviewer 1 Report

 Dear Reviewer,

We appreciate the time you have spent reviewing our paper, we will go carefully answering each of the questions that have arisen.

The paper attempts to assess the link between EEN and rate to remission of CD in pediatric population, and the need for additional therapy (after EEN). This is an interesting and worthy questions considering the burden of CD, and the possible side effect of current alternatives to EEN. Unfortunately the writing style makes it very difficult to adequately understand and interpret the content of the paper. The introduction and methods sections are woefully inadequate, and that makes it very difficulty to critically review the results. The results seem interesting but the lack of information on how the various data was generated makes the interpretation difficult. The authors provided a lot of background in the discussion, which was good. However, they should have committed more time to actually discussing the results presented in the paper.

Introduction:

Line 23: Abstract is incorrectly spelled.

This is very poorly written. The limited information provided in the first paragraph does not support the working hypothesis and the specific goals of the paper. In our opinion, the first paragraph introduces the reader to the topic until it leads to the hypothesis. "Few articles published in Spain, false beliefs about the use of EEN and patients who may not benefit from this therapeutic modality.

Please address the specific concerns listed below directly in the introduction. For such a complex issue, your current  introduction is grossly inadequate and must be substantially revised.

First paragraph: Add a sentence or two to explain what Exclusive Enteral Nutrition means. Done. Thanks.

The introduction needs to be substantially revised to include the following:

Explain EEN to your readers. Thank you for the comment. Done

Provide a basis for the working hypothesis. Present a justification of why the authors think that response to EEN is independent of disease severity, age or the extent of the disease? We said that disease severity measured by wPCDAI because is already known that wPCDAI poorly correlates with mucosal inflammation assessed by SES-CD.

This major concern has not been addressed. There is no evidence presented in the introduction which directly explain the link between EEN and the variables of interest presented in the hypotheses. The paper cited in  the first paragraph indicated that patients with severe disease may not  benefit from EEN. Yet, the working hypothesis states that response to EEN does not depend on severity of flare-up or extent of disease. This makes no sense and is inconsistent with the paper cited. In addition, the authors bring out the issue of age without providing any evidence on the association between age and EEN. This is not a scientific approach for developing hypothesis. Again, the introduction needs substantial revision.

How is severity of flare-up different from extent of the disease? You may have a severe disease confined to terminal ileum or maybe you could have a mild extensive disease (L3L4ab). These are two totally different things.  

Wouldn’t you expect severity of flare-up to correlate positively with severity of disease? In other words, isn’t severity of flare-up also an indicator of the extent of disease? Why are these presented as separate hypothesis since one would be expected to predict the other? Not necessary in my opinion.

Line 54: Provide more information on the clinical features of the patient group which did not respond to. This is on the referenced paper. Reference 3.

The information presented in your paper must be standalone. Sentences must make sense without the need to consult a reference paper. Don’t expect your readers to read one sentence in your paper, and then read another paper in order to understand the context for your study. The fact that you provided a reference is not a reason to leave out important information. This is not how to use references.  Please  address the concern above directly in the introduction.

It is confusing to have a working hypothesis which is very different from the study aim. Thank you for the comment. Our study have 3 aims:

- To determine the rate of remission after induction to remission therapy with EEN

- The predictors of response to EEN (this is on the hand with our hypothesis)

- The need for treatment with biological agents during the first 12 months of the disease.

The hypotheses must flow directly for the aims. Generally, it is advisable to first present the overall aims, and then the specific hypothesis. Also, the only time you talked about biological agents is when you were describing your aim. How do you expect the reader to understand the association between biological agent and EEN if you do not provide any background information to this effect? This is not how you write an introduction to a paper. This introduction must be substantially revised to provide the proper context for your hypotheses.

Revise the introduction to clearly highlight evidence on the factors which predict the benefits (or lack of benefits) of EEN. There is no evidence on that, sorry.

There is plenty of published evidence on factors that potentially predicts the response to EEN. Please do the search well and revised the introduction accordingly.

Materials and Methods:

This section is also very inadequately written. Several important information were not presented. Hence it is impossible to understand the method of data collection.

Provide enough information to help the reader understand why you are referring to this study as a retrospective cohort. Just because you are using pre-exiting data does not necessarily make the study retrospective. We have removed the word cohort to make it easier to read

Removing the word cohort does not make it easier to understand the design your. Instead, it  raise questions about whether the author(s) understand their study design. Clearly indicate  (in the materials and methods section) when outcome and exposure data, including all covariates,  were collected. This is very important.

Indicate whether this was a rolling enrollment. When was the first and last persons enrolled? The investigators were invited to participate in the study in January 2017 and only patients with at least one-year follow up were included.  

Do you mean participants or investigators? Please include a statement about the timeline of enrollment and baseline activities (in the paper).

How long was the follow-up. We analyzed the data after on-year follow-up.

Is it  one year from the first day of enrollment? Or was each person followed for a year? Your responses to these question must be included directly in the paper.

Provide a detailed description of the EEN protocol used in this project: 80% of the centres used Modulen IBD and 20% used another formula during a median duration of 8 weeks (this information is on the text). Only water was allow during EEN period.

The statement you are referring to is actually results. You must include in the materials and methods section the EEN protocol as originally intended.

Provide a detail description of the study/patient population. This information is on table 1.

Table 1 is considered results. You need at least a paragraph in the methods section describing the characteristic of the study population, including the choice of this population, and why they were ideal for this study. Also include information on socio-demographic characteristic of your study participants. This will give a context to your readers, and allow an informed interpretation of your results.

Describe clearly the inclusion and exclusion criteria This is clearly shown in material and methods. Inclusion criteria: paediatric patients diagnosed with CD, based on their clinical, laboratory, endoscopic, radiological and histological criteria, and who were treated with EEN for their first flare-up between January 1, 2014 and December 31, 2016. Exclusion criteria: We excluded patients with ulcerative colitis (UC), those with inflammatory bowel disease unclassified (IBD-U), those undergoing concomitant treatment with steroids or anti-tumour necrosis factor (TNF) during induction with EEN and those who were treated with EEN in successive flare-ups.

Okay, sounds good.

Describe in detail the exact data/samples collected from study participant. Done

What samples did you use to determine the biomarker concentration? Plasma? Serum?   How was whole blood samples collected? Capillary or Venipucture or other means? Please note that biomarker concentration may differ depending on the method of data collection. The responses must be provided directly in the paper.

How were bio samples, stored and analyzed and in which labs? At each Centre

This is not an appropriate response to the question. How were samples moved from point of collection to storage or analytic facility? At what temperature were they stored? How were samples processed?  Which Assays were used to determine the levels of the various markers? This are important in interpreting the various markers.

Provide a description of the informed consent procedures Done

This too was not addressed. When was informed consent obtained, by what means (written, oral etc?) and who administered the informed consent?

Ethical issue.

Provide the official names of all institutions which provided ethical clearance for this study. Already shown on the text “The present study was approved by the ethics committees of the participating centres”

You have not addressed the concern. You need provide the official names of the IRBs which granted ethical clearance. Example..”Ethical clearance who obtained from the Institutional Review Board of the Johns Hopkins University, etc…

Statistical analyses:

The statistical analyses is not at all specific. The authors provided a general description of various statistical tests, without indicating which questions were addressed with the test. This section should be revised to clearly indicate the questions being addressed with each test, and the variables involved. Include a description of how potential confounding was addressed in this paper. If you wish, we can include the statistics used in each table, although in our opinion, so much information makes it difficult to read. We leave it to your discretion.

You need to provide a description of the analyses, referring to specific models and  the variables included in the models. This has to be written as part of the statistical analyses section, and not the tables. You can include a footnote in the tables to reiterate the models used. I don’t think providing a detailed  description of the statistical analyses is “so much information”. It is an important component of the paper.

Provide a clear explanation of how wPCDAI was calculated and how it should be interpreted. We have preferred to reference the wPCDAI calculation to avoid making text too long. wPCDAI interpretation is shown on the Table’s 1 footnote.

wPCDAI is a primary index used in this paper. Your readers must understand how this was generated. The cut-offs are meaningless if readers don’t know how wPCDAI was generated in the first place. Don’t assume that your readers are familiar with this. This should be include in the paper.

How was first flare up assessed and defined? Using wPCDAI . This should be include in the paper, before the results).

Results

The columns in table 1 should have headings. Done

I don’t see that this was done.  The first row in the table should be heading for columns 1 and 2.

All abbreviations in Table 1 should be defined in the footnote Done

All abbreviations in Figure 1 should be defined in the footnote Done

How was IBD diagnosed/defined? Line 63 on the text: diagnosed with CD, based on their clinical, laboratory, endoscopic, radiological and histological criteria.

This statement does not in any way indicate how IBD was diagnosed. Please inclue a clear defition of IBD in the paper.  The definition should clearly indicate who is considered an IBD case, and who is not? What exactly do you mean by criteria. This should be indicated in the paper.

In Table 2, are the P-value generated from uni-variate or multivariate models. If they are from univariate model, also include additional columns to include results from a mulitivariate model. UV and MV are shown on table 3. On table 2 Hypothesis test results are shown.

“Hypothesis test” is meaningless. Are these chi-squares, t-tests etc? Provide a footnote for these tables to clearly explain what tests are used.

In table 3, what other variables were included in the model. Include the results for the variables that were not statistically significant, so that your readers can fully interpret the data. All the variables included on table 2 were included first on the UV analysis, only those variables that presented statistically significant differences or a trend (p<0.15) in the univariate analyses, along with the variables that, based on the theoretical or empirical knowledge, were considered related to the dependent variable and were included on the MV.

Sounds good. Thanks for the clarification. The statement in the statistical analyses section should be revised to include a mention of the variables in Table 2.  Also rovide a footnote to table 3 to indicate the variables that were ultimately adjusted.

Discussion:

The authors provided important information in this section to enable a broader understanding of the problem of pediatric CD. The authors also discussed the acceptability of EEN and possible alternatives to EEN. Thank you for the comment.

The authors spent more time discussing the literature than discussing the actual results presented in the paper. The authors did not adequately discuss why some variables (but not others) were predictors. It appears from table 1 that EEN is not beneficial to the most severe cases (based on WPCDAI and FC values) but this was not discussed. Thank you for this comment, in our opinion we explain the results of our study and also explain the model showed on table 3. “Regarding the predictors to response, although our model is statistically significant, it only partially explains the dependent variable, indicating that additional factors play a role in the response”. We have add the results of Nagelkerke R2 and Cox-Snell R2

The authors mentioned age in their hypothesis, but this was not a focus of their results and discussion. This is shown on table 2 as A1b of Paris. A: age 1b: 10-17 years.

Since age was a variable of interest (as shown in the hypothesis), you have to fully discuss the findings with respect to age.

Thank you.

With best regards
